# Health and Lifestyle of Patients with Mesothelioma: Protocol for the Help-Meso Study

**Leah Taylor [1], Katherine Swainston [2], Christopher Hurst [3,4], Avinash Aujayeb [1,\*] , Hannah Poulter [5] and Lorelle Dismore [3,6]**

1 Respiratory Department, Northumbria Healthcare NHS Foundation Trust, North Tyneside Hospital, Rake Lane, North Shields NE29 8NH, UK
2 School of Psychology, Population and Health Sciences Institute, Faculty of Medical Sciences, Newcastle University, Newcastle NE2 4HH, UK
3 AGE Research Group, Translational and Clinical Research Institute, Faculty of Medical Sciences, Newcastle University, Newcastle NE2 4HH, UK
4 NIHR Newcastle Biomedical Research Centre, Newcastle University and Newcastle upon Tyne NHS Foundation Trust, Newcastle upon Tyne NE4 5PL, UK
5 School of Social Sciences, Humanities and Law, Teesside University, Middlesbrough TS1 3BX, UK
6 Innovation, Research and Development, Northumbria Healthcare NHS Foundation Trust, North Tyneside Hospital, Rake Lane, North Shields NE29 8NH, UK
\* Correspondence: avinash.aujayeb@northumbria-healthcare.nhs.uk or avinash.aujayeb@nhct.nhs.uk; Tel.: +44-0191-293-4087

**Abstract:** Patients with mesothelioma (PwM) have a poor prognosis and are at risk of adverse health outcomes and poor health-related quality of life. Sarcopenia and malnutrition are important prognostic factors for cancer patients and can be partially reversed with adequate nutrition and physical activity/exercise. There is a limited evidence base about the nutritional status of PwM, the understanding of which might potentially influence interventions in PwM. The primary aim of the Help-Meso (Health and Lifestyle of PwM) study is to describe the nutrition, appetite, physical activity and attitude towards lifestyle interventions of PwM. Patients, informal carers and health professionals will be invited to participate in semi-structured interviews and thematic analysis will be performed. The secondary aim of Help-Meso is to assess the feasibility of nutritional screening of PwM via a validated quantitative tool (Malnutrition Universal Screening Tool). The findings from this study will provide an understanding of the health and lifestyle of PwM and the corresponding attitudes of their informal carers and healthcare providers. This information will inform the design of future targeted interventions to improve the nutrition, quality of life of PwM and outcomes. The study has Research Ethics Committee (REC) and Health Research Authority approvals obtained from Wales REC7 (Integrated Research Application System (IRAS) project ID 287193).

**Keywords:** mesothelioma; appetite; nutrition; physical activity

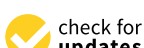



## 1. Introduction

Sarcopenia, cachexia and malnutrition are common in cancer patients [1] and are associated with a range of adverse outcomes, including reduced tolerance to treatments, impaired quality of life, increased mortality and loss of appetite [2,3]. Common symptoms reported by patients with mesothelioma (PwM) include appetite loss, decreased food intake and weight loss [4,5]. Correspondingly, sarcopenia, malnutrition and low body mass index are common in PwM [1,6] and are associated with shorter survival time post-surgery and poor physical activity levels [7].

Malignant mesothelioma is a rare cancer that can develop in the pleura or the peritoneum [8]. Exposure to asbestos is the major cause of malignant pleural mesothelioma, with up to 85% of cases in the United Kingdom (UK) [9] being attributed to it. The use of asbestos was banned in the UK in 1999. However, the latency period between first exposure

and subsequent development of disease is approximately 30–40 years. The UK is probably past its peak of mesothelioma, but with asbestos use unregulated and increasing in some parts of the world, cases will still rise for many years to come. Psychological, physical and nutritional interventions are associated with improved outcomes in patients with cancer and have been extensively studied in patients with lung cancer [10]. These have been incorporated in pre-habilitation programs. Prognosis for mesothelioma is generally poor, with a median survival of 9.5 months and 1-year and 3-year survival rates of 41% and 12%, respectively [9,11]. Interventions that improve physical activity or muscle mass could also benefit PwM [12], but none have been studied prospectively and/or validated. As Muruganandan et al. describe in their review, data from lung cancer trials have been extrapolated to PwM, but PwM have not been studied exclusively. Research in advanced lung cancer also suggests that whilst patients recognize the importance of exercise, nutrition and lifestyle interventions, most do not engage with any change in behavior to improve those aspects [12]. However, there is no qualitative research exploring the experiences of appetite and physical activity of PwM and attitudes towards lifestyle interventions for PwM. Identifying potentially modifiable behaviors could develop targets for intervention. Currently, there are no provision of suitable nutritional and physical activity interventions for PwM, and malnutrition screening is not part of standardized care for PwM.

Thus, the Help-Meso (Health and Lifestyle of PwM) study has been designed. The experiences described above are the result of a complex interplay between healthcare practitioners, patients and their caregivers. This has not been formally elucidated before. Patients and their informal carers may have different needs throughout the mesothelioma journey, including preferences regarding support provision, and clinicians responsible for cancer patients are encouraged to support the integration of malnutrition screening in patients' care. However, patient experience and healthcare professionals' assessments of that experience may not always align [13]. Understanding the patient experience, the carers' perspective and the role of the health professional is invaluable in influencing the design of healthcare services and informing effective interventions. This would provide important insights whilst identifying and addressing specific needs [13,14].

Regular nutritional screening and intervention are recommended in all patients receiving anticancer treatment and in those with an expected survival of more than a few months [15]. Specific tools for use in patients with mesothelioma have been developed [16] but are not widely used. The Malnutrition Universal Screening Tool (MUST) score is a five-step screening adjunct that can help identify malnourished, obese or undernourished (at risk of malnutrition) adults [17]. The MUST score is used in conjunction with management guidelines to develop a care plan related to that individual. It is for use in all settings and by all care workers. The MUST score has not been tested in PwM, and feasibility of the MUST score needs to be attempted.

The development of the MUST score has been described before in a widely cited document [18]. THE 'MUST' REPORT, Nutritional screening of adults: a multidisciplinary responsibility—Development and use of the 'Malnutrition Universal Screening Tool' ('MUST') for adults, was developed in 2003, as there were no reliable tools to detect malnutrition, and there was a prescient need to develop a new one using the current height and weight of patients (to calculate the body mass index) and corroborating unplanned weight loss in the last 3 to 6 months and the effect of any acute illness. It showed good correlational validity when compared to other tools, had predictive validity in hospital and community settings and had high reliability agreement (kappa value ranging from 0.90 to 1.00) [18].

The Help-Meso study therefore aims to (1) describe the experiences of appetite and physical activity of PwM and attitudes towards lifestyle interventions for PwM, with views sought from PwM, their informal carers and health professionals and (2) to assess the feasibility of nutritional screening in PwM using the MUST.

## 2. Materials and Methods

### 2.1. Study Design

Help-Meso is a mixed methods study involving semi-structured interviews and nutritional screening. Patients, informal carers and health professionals will be invited to take part in semi-structured interviews. Patients will undergo nutritional screening using the MUST tool.

### Aims and Objectives

(1) To describe the experiences of appetite and physical activity of PwM and attitudes towards lifestyle interventions for PwM, with views sought from patients, their informal carers and health professionals.

(2) To assess the feasibility of performing nutritional screening in PwM using the MUST. This data will be used to describe the percent of patients who are at risk of malnutrition and for whom a referral for nutritional intervention and assessment is made.

### 2.2. Participants

Participants will be those with mesothelioma diagnosed within Northumbria Health-Care National Health Service (NHS) Foundation Trust, a district general hospital in the northeast of England. Informal carers of those patients will be also invited to take part in the study. Health professionals will include those with a specialist interest and expertise in lung cancer and/or mesothelioma and will include nurses, consultants and doctors in training. Inclusion criteria are shown in Table 1. Those who are unable to provide informed consent, as well as any patient identified by the clinician as inappropriate to approach, for example, too unwell to participate, will be excluded.

**Table 1.** Study inclusion criteria.

| Inclusion Criteria for Each Participant Group |
|---|
| **Patients** |
| • Patients aged ≥18 years diagnosed with mesothelioma. <br> • Able to provide informed consent. <br> • Willingness to take part in either the qualitative interviews (appetite and/or physical activity) and/or nutritional screening. |
| **Informal carers** |
| • Informal carers (relative or friend) aged ≥18 years of PwM. <br> • Able to provide informed consent. <br> • Willingness to take part in an interview. |
| **Health professionals** |
| • Health professional working within the relevant field (i.e., cancer, mesothelioma and thoracic medicine). <br> • Able to provide informed consent. |

#### 2.2.1. Sampling Technique and Identification of Participants

Patients will be screened and invited to participate by a member of the research team. Following eligibility confirmation, the research team will explain the study in detail to the potential participant(s), and if they are happy to proceed, consent will be obtained. Participants will have the opportunity to ask questions and be given time to consider their participation in the study. For those participants requiring more time, the team will provide a study invitation pack that will include a participant information sheet and invitation letter.

Health professionals working in the relevant fields will be identified by the research team. They will be asked to take part in a semi-structured interview and provided with a participant information sheet.

#### 2.2.2. Consent

Informed consent will be gained using one of the following methods: (1) face-to-face consent at a clinical appointment; (2) witnessed consent; (3) postal consent, to minimize participant burden or on the day of the interview. Participants will have the option for the semi-structured interviews to be conducted in their own home. Health professionals have the option for face-to-face consent or witnessed consent, depending on their preference and location.

*Witnessed consent:* The researcher will read the statements on the consent form to the participant for their verbal agreement with the participant, the researcher, and a witness for the researcher present. The consent form will be signed by the researcher and the witness.

*Postal consent*: the researcher will post a consent form to the participant, and a call will be scheduled in for the following week. The researcher will talk the patient through the consent form and the form will be signed by the patient. The researcher will then document the patient's name, date and time of the consent call. The patient will return the original completed consent form by post to be signed by a member of the research team.

#### 2.3. Study Interventions

2.3.1. Qualitative Data: Interviews with Patients with Mesothelioma and Informal Carers

Participants will participate in two semi-structured interviews. The first semi-structured interview aims to understand the experiences of, and influences on, appetite and diet, strategies to maintain adequate nutrition and attitudes towards health and lifestyle interventions. For example, *'can you tell me about your appetite?'*, *'what strategies (if any), have you and your family tried to make sure you are eating enough?'* and *'how would you feel if you were referred for specialist nutritional support?'.* The second semi-structured interview aims to understand experiences of, and influences on, physical activity, strategies to maintain adequate activity levels and attitudes towards lifestyle interventions. For example, *'can you describe how physical active you are?'*, *'what do you currently do to try and keep physically active?'* and *'how would you feel if you were asked to take part in physical activity interventions, such as increased walking, and would there be any benefits or barriers of this?'.*

Informal carers will be invited to take part in the same interview, or they will have the option for a separate interview.

Interviews will take place within patients and carers' own home or remotely (via a video call or telephone call), depending on their preference.

#### 2.3.2. Qualitative Data: Interviews with Health Professionals

Health professionals will be invited to participate in one semi-structured interview. This will focus on their experience of nutrition, weight loss and physical activity of PwM. The interview aims to understand health professionals' current practice in assessing nutritional needs of patients and use of nutritional interventions. For example, *'what is your current practice in assessing nutritional needs of patients with mesothelioma?'* and *who do you think is best placed to assess diet and appetite in patients with mesothelioma?'.* Interviews will take place in hospital or remotely (via a video call), depending on location and individual preference.

Qualitative interviews will provide a rich and detailed account of participants' experiences and how the disease impacts on health and lifestyle. Interviews with carers may provide different insights into the appetite and physical activity behaviors of their relative. Interviews with health professionals will provide invaluable information on how best to support PwM and implement appropriate interventions. We will aim to recruit until data saturation or until a maximum of 20 interviews in each participant group. Interview schedules can be found in Appendix A. Interviews will be audio-recorded and destroyed once transcribed verbatim. Patients' demographic details will be obtained along with clinically relevant data, i.e., diagnosis and treatment. A summary of the study assessments are described in Table 2:

**Table 2.** Study assessments.

| Patients | | |
|---|---|---|
| 1 | A semi-structured interview to understand experiences of, and influences on, appetite, including strategies to maintain adequate food intake and attitudes toward health and lifestyle interventions. | |
| 2 | A semi-structured interview to understand experiences of, and influences on, physical activity, including strategies to maintain adequate activity levels and attitudes toward lifestyle interventions. | |
| 3 | Nutritional screening using the MUST tool (repeated following the guidelines). | |
| 4 | A semi-structured interview following identification of change in nutritional status as identified via MUST scores. Semi-structured interviews will explore experiences of, and influences on, appetite and diet. | |
| **Informal Carers** | | |
| 1 | A semi-structured interview to understand experiences of, and influences on, appetite, including strategies to maintain adequate food intake and attitudes toward health and lifestyle interventions. | |
| 2 | A semi-structured interview to understand experiences of, and influences on, physical activity, including strategies to maintain adequate activity levels and attitudes toward lifestyle interventions. | |
| 3 | An optional semi-structured interview following identification of their relative's change in nutritional status as identified via MUST scores. Semi-structured interviews will explore experiences of, and influences on, appetite and diet. | |
| **Health Professionals** | | |
| 1 | A semi-structured interview to understand experiences of nutrition, weight loss and physical activity of PwM. The semi-structured interview aims to understand health professionals' current practice in assessing nutritional needs of patients and use of appropriate interventions. | |

### 2.3.3. Nutritional Screening

To assess the feasibility of nutritional screening of PwM, participants will complete the MUST with help from a member of the research team. The MUST is a five-step screening tool to identify adults who are malnourished, at risk of malnutrition (undernutrition) or obese (Appendix B). The guidelines recommended by the British Association for Parenteral and Enteral Nutrition (BAPEN) for the collection of MUST data will be followed [15]. This is a quick assessment tool used in many healthcare settings to determine the risk of malnutrition. Information gathered to undertake the MUST tool includes height and weight to determine body mass index (BMI), percentage of unplanned weight loss in three to six months and acute disease score. Scores are calculated to give a low, medium or high risk of malnutrition. For each risk category, there is a management guideline that identifies when repeat screening and referral to dietetic services should take place. We will adhere to the guidelines to ensure patients receive subsequent appropriate care. Patients who score two or above will be referred to nutrition services for a comprehensive assessment and intervention plan. Patients do not have to consent to referral to a dietician, but if they are assessed as being high risk of malnutrition, it will be advised by the patients' healthcare team as part of patients' ongoing care. The research team have received training from the dietetics service to use the MUST.

### 2.4. Data Analysis

All interviews will be analyzed using thematic analysis [19] to allow for an accessible and theoretically flexible approach and a rich and detailed yet complex account of data. An inductive approach will be chosen, with the emergent themes being grounded within the data. Steps include familiarization with the data by reading and re-reading the interview transcripts. Initial codes will be generated through identifying interesting aspects of the

data. Codes will be reviewed and refined, and connections will be identified and clustered together. Codes will be established and organized into potential themes with supporting quotations. The themes will be reviewed and defined and written for publication. Data captured from the nutritional screening will be reported descriptively.

*2.5. Approvals*

Ethical approval has been granted by the Research Ethics Committee (REC) Wales REC7 (Integrated Research Application System (IRAS) project ID 287193) prior to data collection in an NHS secondary care setting, and Caldicott approval (C4127) has been granted from Northumbria Healthcare NHS Trust.

2.5.1. Ethical Considerations

We will offer a flexible approach to data collection. If participants do not want to be interviewed at home, we will conduct the assessments remotely. We appreciate patients giving up their time to participate and can arrange the interviews to be conducted at a time convenient to them. Patients may wish to be supported during the interview/assessment by an informal carer. Consent can be withdrawn at any time. Data collected up to that point of withdrawal will still be used. Participants will be notified of this in the participant information sheet.

Any patient unable to provide consent due to cognitive impairment will not be considered for inclusion in the study.

All researchers who have direct contact with participants will have each necessary approval in place, including research passport, health checks, employment contracts, disclosure and barring service checks, and will be fully trained to undertake all aspects of the study.

2.5.2. Adverse Events

We envisage no risk to the patients; however, if new issues are identified during the study, the research team will notify the clinical team. If any participants were to disclose risk to themselves or others, as well as possible safeguarding concerns, the relevant risk management procedure specific to Northumbria Healthcare NHS Trust would be implemented. The participants will be made aware that their information and data will be kept anonymous; however, if they were to disclose any harm to themselves or others, information will be shared amongst professionals and the correct support offered to the participant.

2.5.3. Data Handling and Storage

The participants' names will only be available to the research team and will not be used in any scientific reports. The data will not be shared with third parties. The consent forms will be stored in a lockable cabinet, separate from any results of the study. Electronic data will be stored on a password-protected, trusted computer server with data encryption and automated back-up. All patients will be assigned a pseudonym to protect their identity and maintain confidentiality, and the wellbeing and safeguarding of individuals are paramount.

## 3. Results

The findings will be published in peer-reviewed journals and presented at national and/or international conferences.

## 4. Conclusions

This protocol describes the design of a mixed-methods study utilizing semi-structured interviews and nutritional screening to explore the health and lifestyle of PwM. Views will be sought from PwM, their informal carers and health professionals. There is a gap in the literature concerning these specific experiences of this group.

Mesothelioma is a complex disease that causes significant morbidity and mortality. Nutritional and physical activity interventions that aim to reverse sarcopenia may improve patients' quality of life. This study will provide important insights into the experiences of appetite and physical activity in PwM to guide the development of appropriate health and lifestyle interventions. Areas for future research also include the ability of the MUST score to detect change in nutritional status in PwM.

**Author Contributions:** L.D., C.H., L.T. and A.A. conceptualized the study and methodology. L.D., K.S., L.T. and H.P. supported obtaining ethical approvals. L.T. and A.A. will identify eligible participants, and L.D., L.T., A.A. and H.P. will consent the participants into the study. L.D., L.T., K.S. and H.P. are responsible for conducting, transcribing and interpreting the qualitative research findings. L.T. and the mesothelioma clinical nurse specialist will perform the MUST, and K.S. will support analysis. All authors have read and agreed to the published version of the manuscript.

**Funding:** This research is funded by Mesothelioma UK Mesothelioma UK | Supporting people with this asbestos cancer 235 Loughborough Road, Mountsorrel, Loughborough, Leicestershire, LE12 7AS.

**Institutional Review Board Statement:** This research will be conducted according to the guidelines of the Declaration of Helsinki and has been approved by an Ethics Committee.

**Informed Consent Statement:** Informed consent will be obtained from all subjects involved in the study.

**Data Availability Statement:** Due to maintaining confidentiality of the participants, the interview transcripts will not be made publicly available.

**Acknowledgments:** The authors would like to acknowledge the patients, informal carers and health professionals who will be taking the time to participate in this research and share their experiences.

**Conflicts of Interest:** The authors declare no conflict of interest.

**Appendix A**

---

**Interview Schedules**

---

**Title: What are health professionals experiences of weight loss, appetite and physical activity in patients with mesothelioma?**

- What is your current practice in assessing nutritional needs of patients?
- What are your expectations of diet and appetite in this patient group?
- What tools do you use to recognise potential problems with diet and appetite?
- Do you routinely discuss diet and appetite with this patient group if in your care? (how do you approach this?)
- What do you think are the potential causes of problems with diet and appetite?
- Do you routinely refer patients to dietetic services?
- When should patients be referred to dietetic services?
- Are there any other interventions you have used in the past?
- Are you aware of any resources they can direct patient to (written/online etc)?
- Who (HCP) do they think is best placed to assess diet and appetite in PwM?
- What are your thoughts about regular nutritional screening in PwM?
- How might diet and appetite impact on other areas of life for PwM?
- Can you tell me what your thoughts are on how physically active you view this patient group?
- Do you see any potential to improve the physical activity levels of this group? (what are the barriers/concerns and enablers).
- Can you suggest any useful ways of promoting increased physical activity in this group?
- How does diet and appetite impact on carers?

---

**Interview Schedules**

**Title: What are patients/carers experiences of appetite?**

- When I say appetite what does that mean to you?
- How would you describe your overall diet/what foods do you prefer to eat?
- Can you tell me about your appetite? (When did you notice a change and why do you think this is?)
- What do you feel influences your appetite?
- What influences your diet?
- What do you think the benefits are of making sure you are eating enough?
- How much thought do you give to your appetite?
- What makes you feel hungry/what reduces this?
- Has your appetite changed in what you opt to eat?
- Can you describe what you are thinking or feeling leading up to a meal?
- How do you feel about your current level of appetite? (what you like to improve this or not?)
- How does mesothelioma impact on your appetite?
- Can you describe a typical day of when you eat/mealtimes? (fixed/flexible) has this changed?
- Who do you usually eat with?
- Can you describe how pleasant or unpleasant the eating experience is to you?
- Can you describe your general mood and well-being? Does this affect your appetite in any way?
- How does diet and appetite impact on you and your family?
- What strategies have you and your family tried to ensure you are eating enough?
- How would you feel if a healthcare professional was to assess your nutritional status?
- How would you feel if you were referred for specialist nutritional advice by a dietician?
- Where would you look to for advice about nutrition and diet? i.e., resources
- We are interested in helping people making sure they are eating enough when living with mesothelioma. We are looking for ways in the future to support this. What are your thoughts if you were asked to;
  - Undertake nutritional counselling (barriers/concerns and enablers for this)
  - Use supplements i.e., a drink or oral tablets (barriers/concerns and enablers for this)
  - Use a specific diet such as increasing your daily food intake? (barriers/concerns and enablers for this)
- What would your preferred method be?
- Can you suggest any other strategies or interventions that would help others with poor appetite?

**Title: What are patients/carers experiences of physical activity?**

- When I say physical activity what does that mean to you?
- What do you think the benefits are of making sure you are physically active?
- How would you describe your overall level of physical activity at the moment?
- How do you feel about your current level of physical activity? (would you like to improve this or not?)
- Were there any times during your illness that you noticed changes in your levels of physical activity?
- How much thought do you give to physical activity?
- Can you describe what you are thinking or feeling leading up being physically active?
- How does mesothelioma impact on your ability to be physically active?
- Can you describe a typical day in terms of your physical activity? has this changed?
- Who are you usually physically active with or is this something you do on your own?
- Can you describe how pleasant or unpleasant being physically active is for you?
- Can you describe your general mood and well-being? Does this affect your ability to be physically active in any way?
- How does physically activity impact on you and your family?
- What strategies have you and your family tried to ensure you being physically active?
- How would you feel if a healthcare professional was to assess your levels of physical activity?
- How would you feel if you were asked to increase your physical activity levels? (if you would be happy to do this, what would you prefer to do i.e., walking, exercise such as swimming etc)
- If you were asked to take part in a physical activity intervention how would you feel about this? (i.e., being asked to walk more) what would the barriers/concerns/enablers be?
- Where would you look to for advice about physically activity? i.e., resources
- How did COVID-19 pandemic affect your levels of physical activity?



**Appendix B**

The Malnutrition Universal Screening Tool (MUST) (reproduced from https://www.
bfwh.nhs.uk/our-services/community-nutrition-and-dietetics/nutritional-support/, ac-
cessed on 1 July 2022).

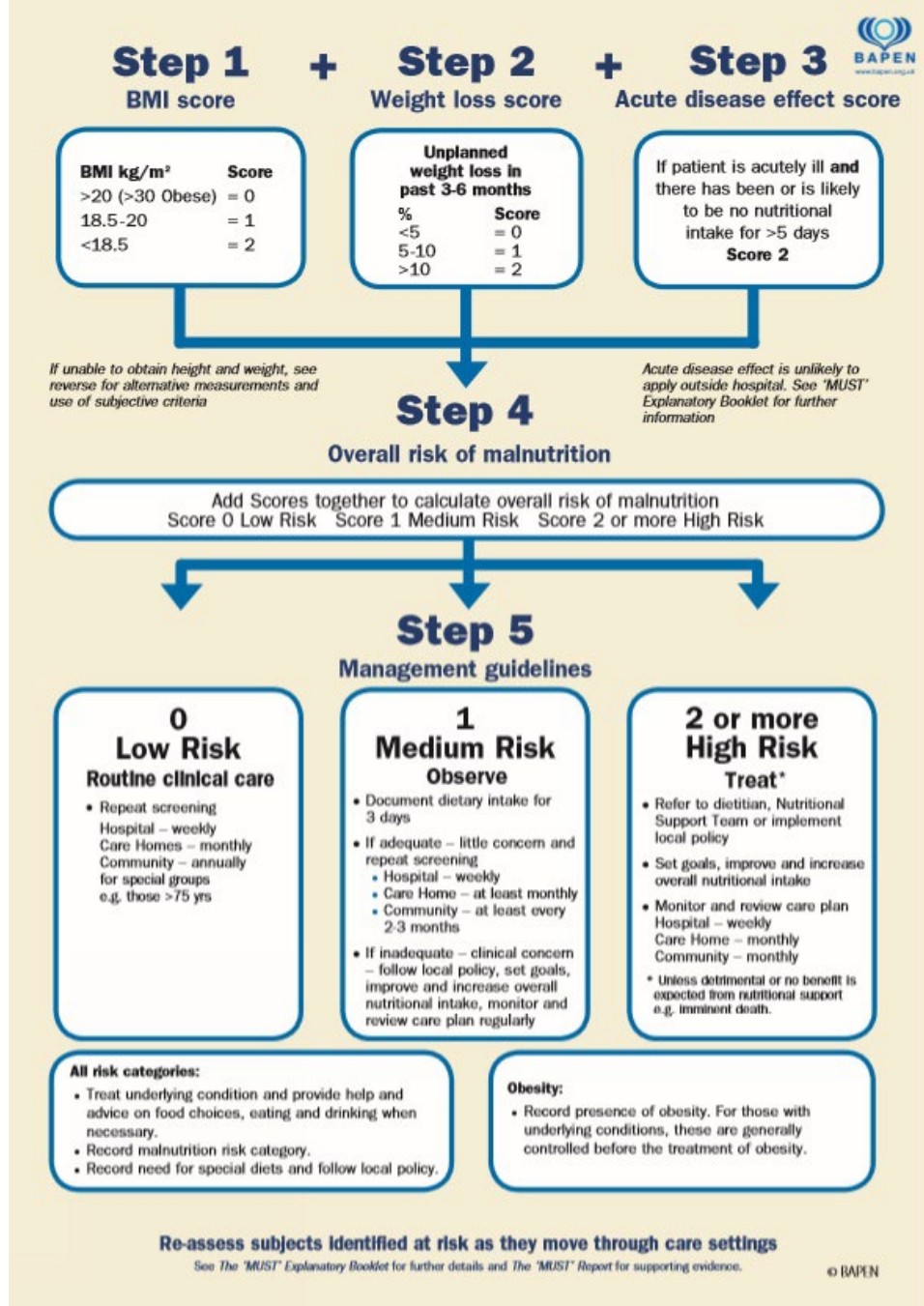

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
