# Peer review of "Health and Lifestyle of Patients with Mesothelioma: Protocol for the Help-Meso Study"

_2673-527X, doi:10.3390/jor2030011_

Round 1
Reviewer 1 Report
I appreciated the proposal of an evaluation of nutritional status in mesothelioma patients.
I have just a question for Authors: is it possibile that the chance to choose to carry out the interview either in presence or in videoconference could lead to a distortion of results? An interview in presence could be more precise and stimulating for patients that a video one.
Minor: I think that the acronim PwM should be explained in the abstract
Author Response
Thank you for your comments and we will try explain this now
We favour an interview in presence over a videoconference however, we have offered a flexible approach for our participants to increase participation in the research. We anticipate that the majority of patients and carers will be interviewed face to face.
The acronym PwM has been included in the first sentence of the abstract following Patients with mesothelioma.
Reviewer 2 Report
Taylor et al proposed a manuscript protocol entitled : "Health and Lifestyle of patients with mesothelioma: protocol 2 for the Help-Meso study".
In general, the protocol is well described with aim and adequate methods. The first endpoint is to describe the experiences of appetite and physical activity of PwM and attitudes towards lifestyle interventions for PwM. Secondary endpoint is to assess the feasibility of performing nutritional screening in PwM using the MUST questionnary, to evaluate the risk of malnutrition. This study is of great interest : nutrition, nursing and supportive care broadly impact survival and quality of life for patients with mesothelioma.
3 participant groups are planned : Patients, informal carers and Health professionals.
Study interventions will include
Only minor points need modifications :
- interviews with patients with mesothelioma and informal carers
- interviews with health professional
- Nutritional Screening (MUST)
- line 4 : please check "', , "
- Abstract : please define "PwM" in the abstract
- Please check typo for ref 9 "Woolhouse"
This protocole paper can be consider for publication.
Thanks to authors and editorial team for this review request
Author Response
|
Thank you for your comments and we have highlighted the 2 points which we will try address- I think there is a slight typographical error and we think that only the minor points need modifications
"PwM" has been defined in the abstract.
Woolhouse no longer underlined.
|
Reviewer 3 Report
The project summarized in the article is very interesting and deserve to be investigated in the next future.
Despite that, there is a major drawback in the developing of the questionnaire that should be addressed.
Minor points:
In the introduction section, the Authors state that "Prognosis for mesothelioma is generally poor and interventions that improve physical activity or muscle mass could also benefit PwM".
Please clarify and provide references.
Major point
The Authors should summarize the methods used in the development of the questionnaire that they want to use.
If a new questionnaire is to be developed, testing will establish that it measures what is intended to be measured, and that it does so reliably (10.1186/1745-6215-11-2). The validity of a questionnaire may be assessed in a reliability study that assesses the agreement (or correlation) between the outcome measured using the questionnaire with that measured using the 'gold standard'. However, this will not be possible if there is no recognised gold standard measurement for outcome. The reliability of a questionnaire may be assessed by quantifying the strength of agreement between the outcomes measured using the questionnaire on the same patients at different times. If new questions are to be developed, the reading ease of the questions can be assessed using the Flesch reading ease score. This score assesses the number of words in sentences, and the number syllables in words. Higher Flesch reading scores indicate material that is easier to read (10.1037/h0062427).
Author Response
Thank you for your kind comment and we will try address the points that you have raised
Minor points: Introduction has been updated with relevant references- please see text.
Major point: We are keen to do a feasibility study initially. The study will utilise an existing questionnaire, the MUST, which is already well established to assess nutrition, and accordingly no new questionnaire is to be developed. The study will assess the feasibility and acceptability of patients completing this tool and no changes will be made to the questions within the measure. If this is successful, we then plan to develop a formal reliability study with appropriate confounder correction to see if the MUST score can be applied to mesothelioma patients and capture change over time.
We have added a line to the conclusion to reflect that
Round 2
Reviewer 3 Report
The Authors have not responded my comments.
Author Response
We apologise that we have been clearer in our previous remarks
For the first comment : we have elaborated on this more, and hope that this is now acceptable.
Prognosis for mesothelioma is generally poor, with a median survival of 9.5 months, and 1-year and 3-year survival rates of 41% and 12% respectively [9,11]. Prognosis for mesothelioma is generally poor, with a median survival of 9.5 months, and 1-year and 3-year survival rates of 41% and 12% respectively [9,11].Interventions that improve physical activity or muscle mass could also benefit PwM [12] but none have been studied prospectively and/or validated. As Muruganandan et al describe in their review, data from lung cancer trials have been extrapolated to PwM but PwM have not been studied exclusively. Research in advanced lung cancer also suggests that whilst patients recognize the importance of exercise, nutrition and lifestyle interventions, most do not engage with any change in behaviour to improve those aspects [12].
For the major comment, again we apologise for not being clearer. We are not developing a new questionnaire. The MUST questionnaire was developed in the early 2000's and provides a theoretical and practical framework for the clinical detection and management of nutritionally responsive conditions, caused by
physical and psychosocial problems. It is validated and is internally consistent and reliable. It has very good to excellent reproducibility when different observers assess the same patients in hospitals (inpatients and out-patients), GP surgeries, and care homes (kappa values between 0.8 and 1.0)
It has not been tested in patients with mesothelioma and we are thus just testing the feasibility of this tool in such patients and in our text, address how we will assess the feasibility of this.
must-report.pdf (bapen.org.uk) has been cited over 400 times, and provides the basis of the MUST score. A new parapragh detailing that has been done (lines 82-91)
Round 3
Reviewer 3 Report
I am sorry but maybe I have not been clear in the previous comments.
The "new" questionnaire that the Authors should validate in a sound and scientific manner is not MUST but the Appendix A that reports several questions for Clinicians, Patients and caregivers.